# Rh-induced Support Transformation and Rh Incorporation in Titanate Structures and Their Influence on Catalytic Activity

**János Kiss [1,2,*], András Sápi [1], Mariann Tóth [1], Ákos Kukovecz [1] and Zoltán Kónya [1,2]**

[1] Department of Applied and Environmental Chemistry, University of Szeged, Interdisciplinary Excellence Centre, Rerrich Béla tér 1, H-6720 Szeged, Hungary; sapia@chem.u-szeged.hu (A.S.); tothmaja@gmail.com (M.T.); kakos@chem.u-szeged.hu (Á.K.); konya@chem.u-szeged.hu (Z.K.)

[2] MTA-SZTE Reaction Kinetics and Surface Chemistry Research Group, Rerrich Béla tér 1, H-6720 Szeged, Hungary

[*] Correspondence: jkiss@chem.u-szeged.hu

**Abstract:** Rh is one of the most effective metals in several technologically important heterogeneous catalytic reactions, like the hydrogenation of $CO_2$, and CO, the $CO+H_2O$ reaction, and methane and ethanol transformations. Titania and titanates are among the most frequently studied supports for Rh nanoparticles. The present study demonstrates that the nature of the support has a marked influence on the specific activity. For comparison, the catalytic activity of $TiO_2$ P25 is also presented. It is pointed out that a certain amount of Rh can be stabilized as cation ($Rh^+$) in ion-exchange positions (i.e., in atomic scale distribution) of the titanate framework. This ionic form does not exists on $TiO_2$. We pay distinguished attention not only to the electronic interaction between Rh metal and the titania/titanate support, but also to the Rh-induced phase transitions of one-dimensional titanate nanowires (TiONW) and nanotubes (TiONT). Support transformation phenomena can be observed in Rh-loaded titanates. Rh decorated nanowires transform into the $TiO_2(B)$ phase, whereas their pristine counterparts recrystallize into anatase. The formation of anatase is dominant during the thermal annealing process in both acid-treated and Rh-decorated nanotubes; Rh catalysis this transformation. We demonstrate that the phase transformations and the formation of Rh nanoclusters and incorporated Rh ions affect the conversion and the selectivity of the reactions. The following initial activity order was found in the $CO_2 + H_2$, $CO + H_2O$ and $C_2H_5OH$ decomposition reactions: $Rh/TiO_2$ (Degussa P25) ≥ Rh/TiONW > Rh/TiONT. On the other hand it is remarkable that the hydrogen selectivity in ethanol decomposition was two times higher on Rh/TiONW and Rh/TiO(NT) catalysts than on $Rh/TiO_2$ due to the presence of $Rh^+$ cations incorporated into the framework of the titanate structures.

**Keywords:** Rh catalyst; titanate nanowires; titanate nanotubes; ion-exchange; phase transformation; $CO_2$ hydrogenation; methanation; water-gas-shift reaction; ethanol decomposition

## 1. Introduction

### 1.1. General Surway

The heterogeneous catalysis remains as a very well focused research field in the 21st century. The search for new, effective catalysts is very important for future energy production, energy storage and also from an environmental point of view [1,2]. $CO_2$ is a problem and there is a need for the chemical activation of such stable molecules. Catalytic $CO_2$ hydrogenation not only reduces the anthropogenic emission of $CO_2$ but also produces value-added liquid fuels and feedstock chemicals. $CO_2$ conversion using $H_2$ produced from the electrolysis of water generated by wind or solar energy



produces carbon monoxide (CO), methane ($CH_4$) and methanol ($CH_3OH$), etc. This approach is considered promising to reduce the atmospheric $CO_2$ level [3–10]. Because of the chemical inertness of $CO_2$ and its thermodynamic stability, the chemical activation of $CO_2$ is a rather popular buzzword in contemporary catalysis and surface science. However, efficient and selective $CO_2$ conversion remains a challenge [11–14]. The water gas shift reaction (WGSR) [15,16], ethanol (methanol) decomposition and ethanol steam reforming (ESR) serve hydrogen for the energy source [17–19].

The hydrogenation of $CO_2$, the WGSR and the ethanol transformation reactions, including ESR, were investigated extensively on Rh-related catalysts because of Rh was found to be an excellent catalytic material among noble metals. The supports, depending on their nature, significantly influence the catalytic activity of the Rh. $TiO_2$ was the most effective support and the least impressive one was $SiO_2$. For example, in $CO_2$ hydrogenation using $TiO_2$ support, mainly methane was formed with a high conversion and with a high selectivity, in some cases higher than 95% [20–22]. The catalytic reactions have been investigated as a function of the electric properties of the $TiO_2$ support adjusted by doping $TiO_2$ with lower cations and higher valences [23,24]. It was demonstrated that the electric conductivity of $TiO_2$ influences the catalytic properties of Rh.

A broad literature coverage and excellent reviews are available on $TiO_2$ and $TiO_2$ nanostructures [25–27], perovskites [28] and anodically oxidized vertically oriented freestanding $TiO_2$ nanotube arrays [29,30]. The last 15–20 years have seen a steadily increasing number of studies on the properties of polytitanate-based layered nanostructures like titanate nanowires and nanotubes. Various tubular metal oxides have been developed recently and are of interest because they are expected to exhibit novel physical and chemical properties. One-dimensional $TiO_2$-related nanomaterials with a high morphological specificity, such as nanotubes and nanowires, have attracted considerable attention due to their interesting physicochemical properties [31–34]. Kasuga et al. [35] prepared the first titanate nanotubes (TiONT) by hydrothermal synthesis. Later, this method was applied to convert the self-assembled TiONT into nanowires (TiONW) in a revolving autoclave in our laboratory [36,37]. These 1D nanostructures are of great interest in catalysis because they have high specific areas and cation exchange capacities, providing a high metal (e.g., Co, Cu, Ni, Ag and Au) dispersion [34,38–41]. Bavykin and Walsh published an excellent review about the preparation, characterization and applications (including catalysis) of titania and titanate nanotubes [42]. A comprehensive review was also published specifically about characterization at the atomic scale and the surface properties of metal-modified TiONT and TiONW [43,44].

Recently, it was discovered that titanate nanostructures are able to stabilize Au at a high dispersion [45–51]. To date, several important reactions have been discovered to be catalyzed by titanate-supported gold. It was found that gold-containing TiONT has a higher activity than Degussa P-25 in the photo-oxidation of acetaldehyde [47], in the photo-induced degradation of formic acid [48], in the low-temperature WGS reaction [49] and in carbon monoxide oxidation [46,50,51]. They are also active catalysts of thermally induced $CO_2$ hydrogenation [52]. Very recently, it was revealed that gold nanoparticles supported on titanate nanowires are efficient in the UV photo-induced reaction of methane (with and without water) [13,53] and in $CO_2$ hydrogenation [14]. Additional examples confirmed that other noble metals supported on titanate nanostructures also perform remarkably in certain catalytic processes. Deposited Pt, Pd, Ru, and Au on the surface of titanate nanotubes were prepared for catalytic purposes [54]. Titania and titanate nanostructures and their chemically doped and cocatalyst decorated derivatives have been extensively studied in degrading organic impurities being present water as well as on solid surfaces. While the electrons accumulated typically on the co-catalyst nanoparticles (Pt, Pd, Rh) are expected to interact with unsaturated and aromatic bonds in organic moieties, the holes on the surface of $TiO_2$ are responsible for the initiation of oxidative processes that may result in C–C bond scission, dehydrogenation and the like [55–62]. Recently, platinum catalysts supported on layered protonated titanate-derived titania nanoarrays were found to have a high activity in CO and NO oxidation as compared to Pt catalysts through wet-impregnation on the anatase $TiO_2$ [63].

As Rh on titania-based supports exhibits excellent catalytic activity in many reactions, we intended to test the Rh/titanates in some technologically important reactions. Before testing, it is desirable to summarize the Rh-titanate interactions and review the chemical environment of Rh on titanate nanostructures as these parameters play an important role in heterogeneous catalysis. We should first provide a literature survey on Rh-induced transformations of titanates (wires and tubes) and the surface characterization of Rh on titanate supports, including the formation of Rh nanoparticles on the surface and the incorporation of $Rh^+$ ions into the titanate frameworks.

### 1.2. Literature Review of Phase Transformation of Heat-Treated Pristine (H-Titanate) and Rh-Decorated Titanates

Titanate nanowires (TiONW) and nanotubes (TiONT) were prepared by hydrothermal conversion of anatase $TiO_2$ as described previously [35,36,53]. After preparation, acid washing was applied in order to replace as much $Na^+$ ions in the framework of protons as possible. The resulting material is generally called "H-form" titanate.

A characteristic difference between the behavior of titanate nanotubes and nanowires is that in heat-treated nanotubes, the $E_{2g}$ mode is found at exactly the anatase position (636 cm$^{-1}$) from 573 K onwards, whereas in nanowires, this mode experiences a gradual red shift from 648 cm$^{-1}$ at 573 K to 636 cm$^{-1}$ at 873 K [64–66]. A similar effect was observed by Du et al. [67,68] and Scepanovic et al. [68,69] in their temperature-dependent in situ Raman studies of nanocrystalline anatase.

In the case of Rh-loaded titanates, we concluded from Raman spectra that (i) the heat treatment of Rh-loaded titanate nanostructures yield different phase structure from 673 K, (ii) Rh loaded TiONT transforms into anatase, and (iii) the Rh loaded TiONW exhibits the $TiO_2$(B) structure [69,70].

From the study of XRD, it was concluded that the structure of Rh-free (pristine) TiONW is a mixture of different titanate forms, mostly with $TiO_2$(B) and $H_xNa_{(2-x)}$ components (Figure 1A).The transformation is continuous during the thermal annealing. Around 473 K and 573 K, the layered structure collapses and the anatase phase shows up with a low crystallinity. At a higher temperature, the formation of the anatase phase becomes dominant as the electron diffraction patterns show in Figure 1A, together with the appearance of the characteristic anatase reflections (101), (004), (200), (105), (211) and (204) at 25.3°, 37.8°, 48.1°, 53.9°, 55.1° and 62.4°.

**A**

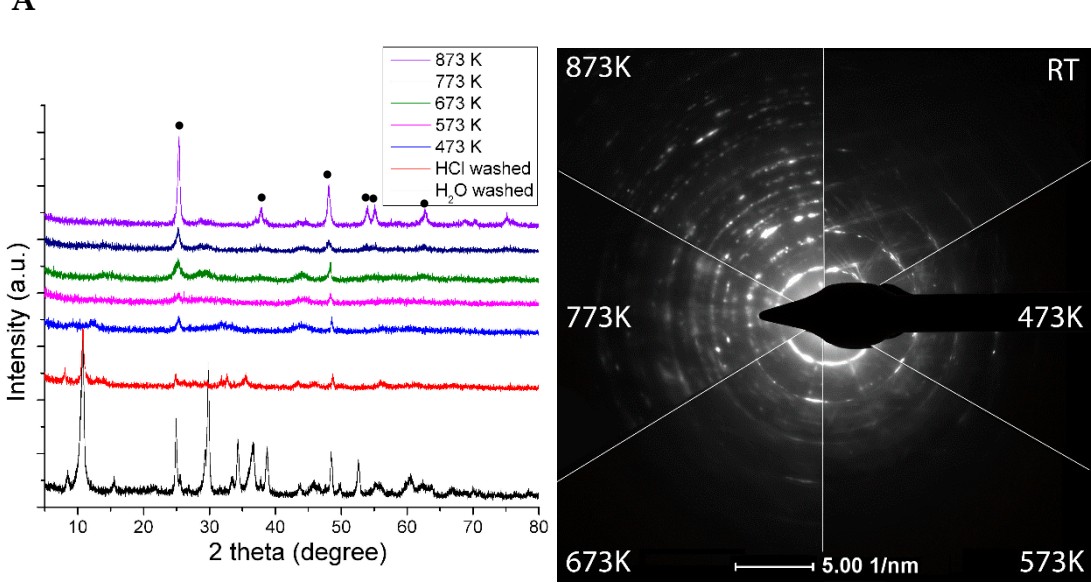

**Figure 1.** *Cont.*

**B**

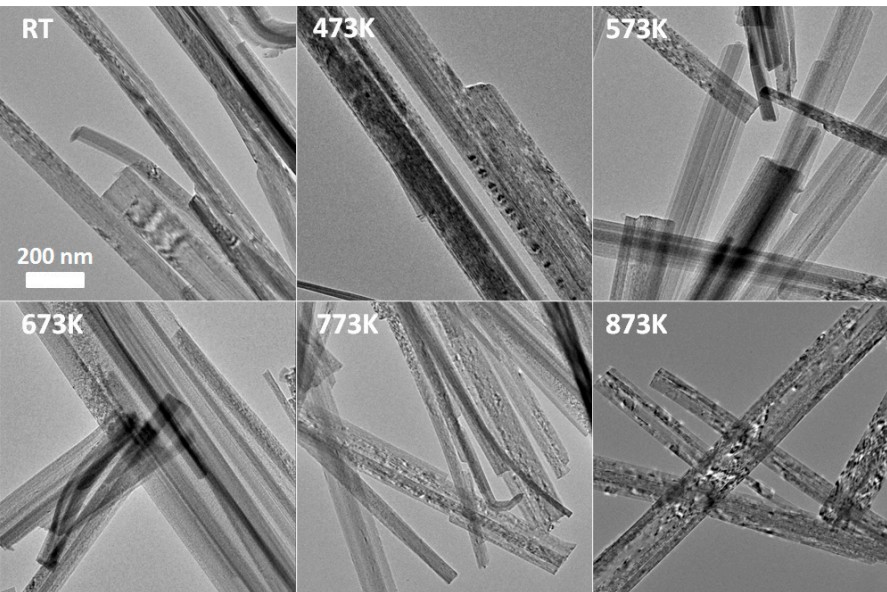

**Figure 1.** (**A**) XRD of TiONW treated at different temperatures. The bottom graph displays the XRD profile of H$_2$O washed TiONW. ● denotes anatase reflection (**B**) TEM images of TiONW treated at different temperatures. Reproduced from [70].

The TiONW preserves the wire-like morphology during the heating up to 873 K. The holey structure is due to the continuous transformation of TiONW to TiO$_2$ (anatase) followed by water desorption from the structure (Figure 1B).

XRD shows that acidic treatment resulted in a degradation of the initial structure of nanotubes (Figure 2A), which demonstrates the disappearance of the reflection belonging to the tubular interlayer distance (2Θ = ~10°). The protonation also catalyzes the transformation of the TiONT to the anatase phase [70]. There is no significant effect on the structure below 673 K; however, at a higher temperature, the anatase formation became dominant, as evidenced by the appearance of the anatase reflections (Figure 2A). At higher temperatures, the increased intensity and lower half-width indicate the improvement of anatase crystallinity, as indicated by the electron diffraction patterns in Figure 2A.

The electronmicroscopic pictures (TEM) in Figure 2B demonstrate the tubular morphology of the as-synthesized TiONT with a diameter of ~7 nm and a length up to 80 nm. In correlation with the XRD results, there is no morphological degradation after heat treatment up to 573 K. The tubular structure collapses and transforms into rod-like nanostructures at a higher temperature. Around 873 K, the tubular morphology is totally collapsed, short nanorods and TiO$_2$ nanoparticles appear with an average size of ~10 nm.

The phase transformation during heat treatment accompanied by structural water loss was investigated and discussed previous studies [31,36,37,42,43,70]. The origin of H$_2$O evolution is the adsorbed or lattice water and the surface reaction between hydroxyl groups and hydrogen during the recrystallization process [43,70]. These processes could significantly increase the number of defects in TiONW and TiONT and this can catalyze the phase transformation of titanates. Heat treatment induces a reduction of Ti$^{4+}$ in titanates to Ti$^{3+}$ and Ti$^{2+}$, but their detection in the surface layers is not always successful due to the fast oxygen transport from bulk to surface. The reduction extent of these cations with the annealing temperature was monitored by treating nanotube samples in situ in inert atmosphere at different temperatures [43]. When the sample was annealed, the population of reduced Ti$^{3+}$ atoms increased, giving a Ti$^{3+}$/Ti$^{4+}$ surface atomic ratio of 0.046 and 0.06 at 573 and 773 K, respectively (Table 1).

**A**

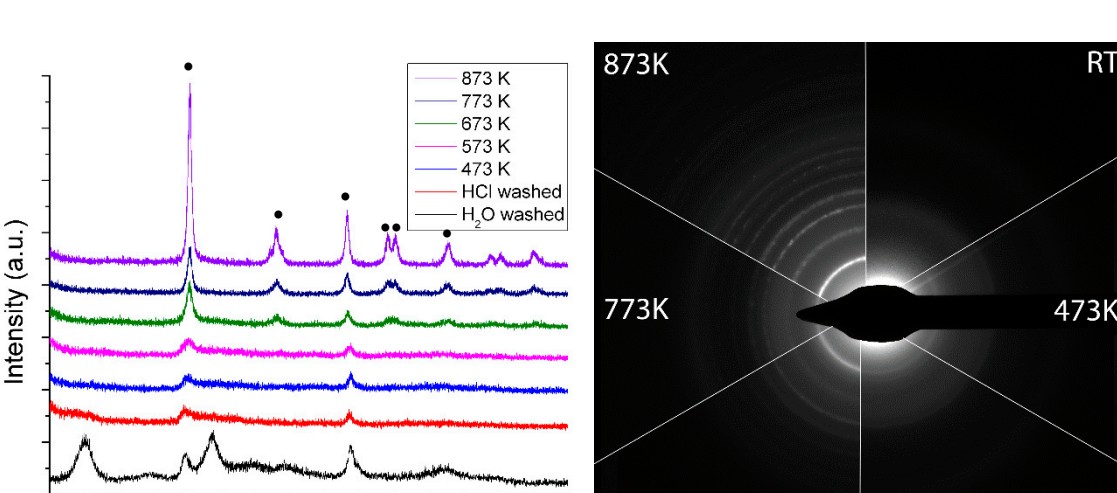

**B**

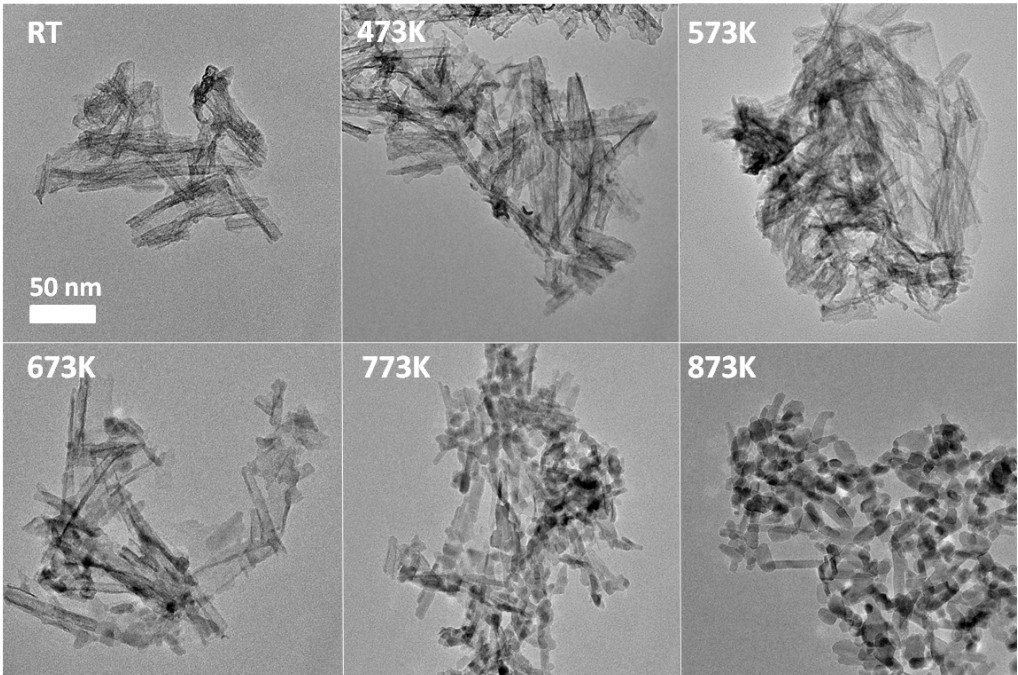

**Figure 2.** (**A**) XRD of "H-form" TiONT as a function of temperature. The bottom curve shows the XRD profile of $H_2O$ washed TiONT. ● denotes anatase reflection. (**B**) TEM images of TiONT treated at different temperatures. Reproduced from Ref. [70].

**Table 1.** XPS parameters of Ti $2p_{3/2}$ and O 1s derived from spectral fitting. Reproduced from [43].

| Annealing Temperature (°C) | Assignment | Binding Energy (eV) | FWHM [a] (eV) | Surface Atomic Ratio $Ti^{3+}/Ti^{4+}$ | Surface Atomic Ratio O/Ti |
|---|---|---|---|---|---|
| 110 | O 1s | 530.8 | 1.3 | 0.026 | 2.48 |
| | $Ti^{3+} 2p_{3/2}$ | 457.5 | 1.2 | | |
| | $Ti^{4+} 2p_{3/2}$ | 459.1 | 1.2 | | |
| 200 | O 1s | 530.8 | 1.2 | 0.048 | 2.17 |
| | $Ti^{3+} 2p_{3/2}$ | 457.8 | 1.2 | | |
| | $Ti^{4+} 2p_{3/2}$ | 459.2 | 1.1 | | |
| 300 | O 1s | 530.8 | 1.2 | 0.046 | 1.96 |
| | $Ti^{3+} 2p_{3/2}$ | 458.1 | 1.4 | | |
| | $Ti^{4+} 2p_{3/2}$ | 459.4 | 1.1 | | |
| 400 | O 1s | 530.9 | 1.2 | 0.045 | 1.89 |
| | $Ti^{3+} 2p_{3/2}$ | 458.1 | 1.4 | | |
| | $Ti^{4+} 2p_{3/2}$ | 459.4 | 1.1 | | |
| 500 | O 1s | 530.8 | 1.2 | 0.060 | 1.97 |
| | $Ti^{3+} 2p_{3/2}$ | 458.1 | 1.6 | | |
| | $Ti^{4+} 2p_{3/2}$ | 459.4 | 1.1 | | |

[a] Full width at half maximum.

Rh/TiONW transform into the $TiO_2(B)$ structure concluded from XRD measurements as opposed to the rhodium-free counterparts' recrystallization to anatase [70]. The dominant reflections attributed to anatase 25.3° (101) and $TiO_2(B)$ 24.9° (110) in the case of Rh-decorated titanate nanowires with a degree of crystallinity of up to 673 K. The lower FWHM values at higher temperatures indicates the fusion of nanoparticles. In the case of Rh-decorated nanotubes (Rh/TiONT), the anatase phase is dominated; reflections are at (101), (004), (200), (105), (211) and (204) at 25.3°, 37.8°, 48.1°, 53.9°, 55.1° and 62.4°, respectively [70].

TEM images of Rh-decorated nanowires (Figure 3A) and nanotubes (Figure 3B) thermally treated at 673 K show the presence of homogeneously dispersed nanoparticles on the surface of the titanate nanostructures [70]. The average nanoparticle diameter is 4.9 ± 1.4 nm and 2.8 ± 0.7 nm in the case of nanowires and nanotubes, respectively, as shown in the corresponding size distributions. The difference in average diameter and distribution broadening can be explained by the differences in the crystal transformation process, as discussed above. Moreover, the surface diffusion and coalescence kinetics of Rh nanoparticles can also be different on tubular and wire-like titanate nanostructures.

A

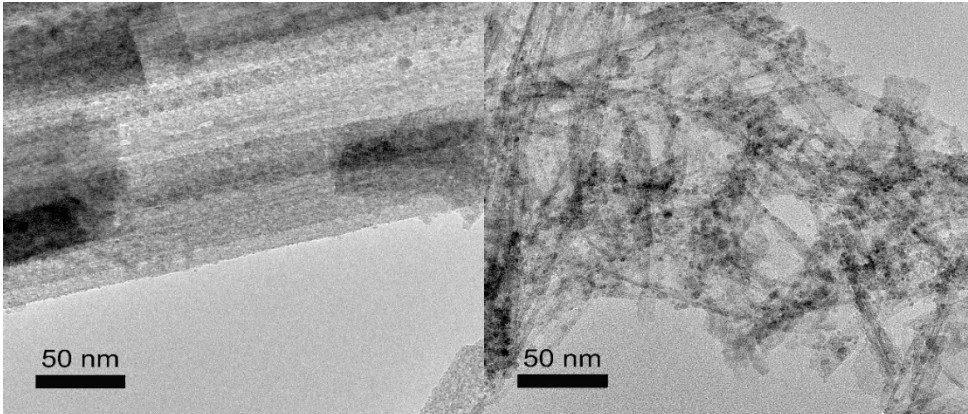

B

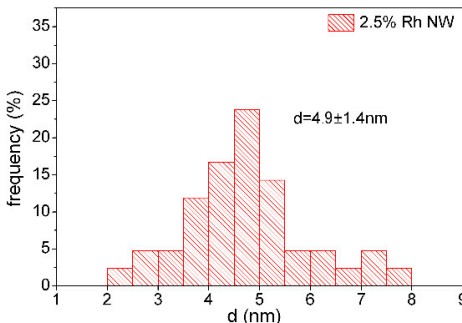
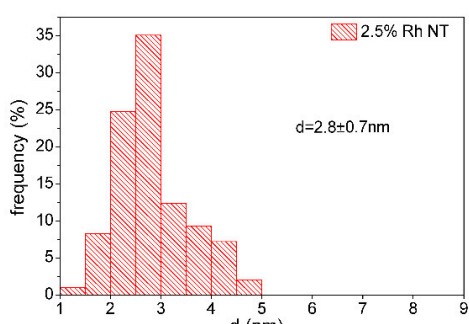

**Figure 3.** Typical TEM images of 2.5% Rh decorated titanate nanowires (**A**) and titanate nanotubes (**B**) thermally annealed at 673 K and the corresponding size distribution of Rh nanoparticles. Reproduced from [70].

It is important to mention that the original size distribution was maintained even at relatively high temperatures (reduction temperature of the catalysts: 473–573 K) on both nanowires and nanotubes. Rh clusters of controlled size can be prepared by physical vapor deposition (PVD) [71–73] and using Rh organometallic precursors [74,75] on $TiO_2$(110) as well. However, STM [71,76], XPS and LEIS [67] experiments revealed that depending on the original cluster size and the evaporation temperature, the agglomeration of Rh nanoparticles can be significant even below 500 K on that surface. Therefore, the relatively small cluster sizes obtained on titanate nanowires and nanotubes may indicate that metal diffusion on these nano objects is limited compared to that on well ordered titania.

*1.3. Summary Results on the Morphology and Chemical State of Rh Nanoparticles on Titanates*

The morphology of Rh supported on titanate nanowires and nanotubes was investigated by FTIR spectroscopy employing adsorbed CO as a probe molecule sensitive to the local surface structure. Adsorbed CO exhibits at least three different stretching frequencies belonging to certain adsorption sites of Rh on oxide supports [77–82]. The band at 2070–2030 $cm^{-1}$ is due to CO adsorbed linearly to $Rh^0$ (depending on the coverage), the band at ~1855 $cm^{-1}$ represents bridge-bonded CO ($Rh_2$–CO) and the features at ~2100 $cm^{-1}$ and at ~2020 $cm^{-1}$ correspond to the symmetric and asymmetric stretching of Rh+$(CO)_2$ (twin CO), respectively. These latter IR signals were detected when the crystallite size was very small [77,81].

On nanowires, the twin form was dominant (2027 and 2097 $cm^{-1}$), the signal corresponding to the linear form between the twin peaks was much smaller and the bridge form was hardly observable (Figure 4A). On nanotubes, the linearly adsorbed CO features showed up at 2075 $cm^{-1}$ between the peaks at 2100 and 2036 $cm^{-1}$ (twin form) (Figure 4B). From these IR studies, we may conclude that a

significant part of Rh exists in small cluster sizes (1–3 nm), probably with the $Rh^+$ oxidation state on both nanowires and tubes in harmony with the XPS results.

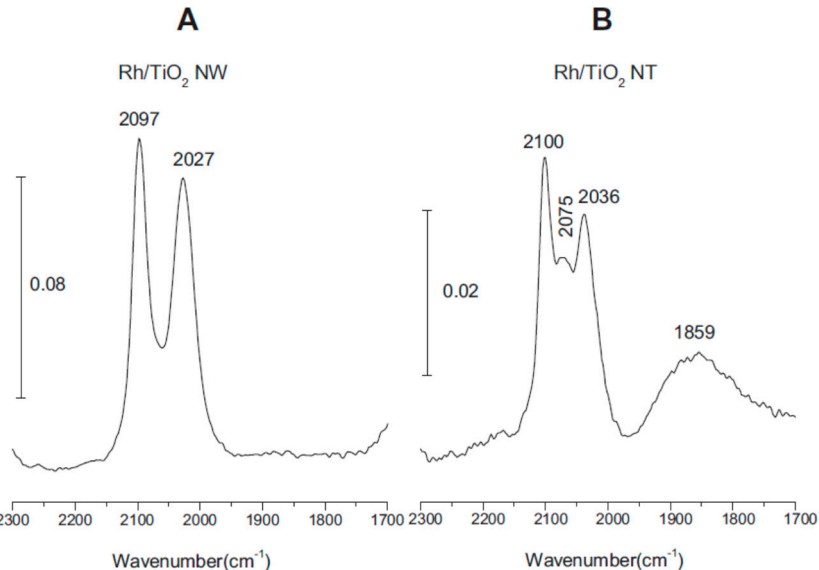

**Figure 4.** Infrared spectra of adsorbed CO at 300 K; (**A**) 1% Rh/TiONW, (**B**) 1% Rh/TiONT.

Figure 5 reveals the binding energies of Rh 3d orbitals in titanate nanowires and nanotubes. The photoemission from the Rh 3d peak centered at 309.3 eV at 1% Rh content and 308.3 eV at 2% metal content clearly suggests the existence of an oxidation state or a different morphology from the bulk, as the metallic Rh photoemission for Rh $3d_{5/2}$ is at 307.1 eV [70]. The XP spectra of Rh 3d for 2% Rh content are presented in Figure 5. The nearly 2 eV shift relative to metallic Rh can be attributed to the width of the nanoparticle distribution. The binding energy is affected by the relaxation energy and this so-called "final-state" effect depends on the particle size [83]. A higher binding energy in XPS may correspond to very small metal particles and Rh ion ($Rh^+$) stabilized in the framework of wires and tubes. It is strongly suggested that the higher binding energy peak corresponds mainly to Rh ion formed in ion-exchange process.

The stabilization of Rh ion and clusters in small size in titanates and their influence on the phase transformation of both titanate formations can be explained by the increased electronic interaction between Rh and titanate structures. A very similar strong electronic interaction was observed in several cases between reduced titania ($TiO_2$) and metals, including Rh [84–86], except ion-exchange possibility. Due to the preparation methods of titanate nanostructures and the mild reduction of Rh/titanates, the nanowires and nanotubes may contain significantly more defects than commercially used reduced titania. The presence of a high number of defects and oxygen vacancies in titanate could initiate an increased electron flow between metal and titanates. On the other hand, ion exchange between protonated titanates and rhodium occurs, forming positively charged Rh, similarly to silver, cobalt and gold on titanates [34,40,41].

In the following section of the review, the focus is on the effect of the structural differences of the titania-based catalyst support as well as the oxidation state and chemical environment of the active metal (Rh) on the industrially and environmentally important catalytic reactions, such as $CO_2$ hydrogenation, CO + $H_2O$ reaction, and $C_2H_5OH$ decomposition. Nanostructured titanates are characterized by a relatively high specific surface area. The high specific surface area of the support facilitates the high dispersion of the catalyst, while the open mesoporous make the efficient transport of both reagents and products possible [42,43]. In addition to the high surface areas, the titanates contain huge number of defects which also play a significant role in the catalytic reaction. Protonated nanowires and nanotubes have good ionic exchange properties. The incorporation of the ionic form of

the metal precursor ($Rh^+$) to the structure can significantly help to increase the loading of the catalysts and maintain a high catalyst dispersion during the reactions, as was nicely presented in the case of platinum supported on layered protonated titanate nanowires [63]. The metal cations resulted in a strong interaction between metal and titanate support, leading the enhanced thermal and chemical stability of the catalyst.

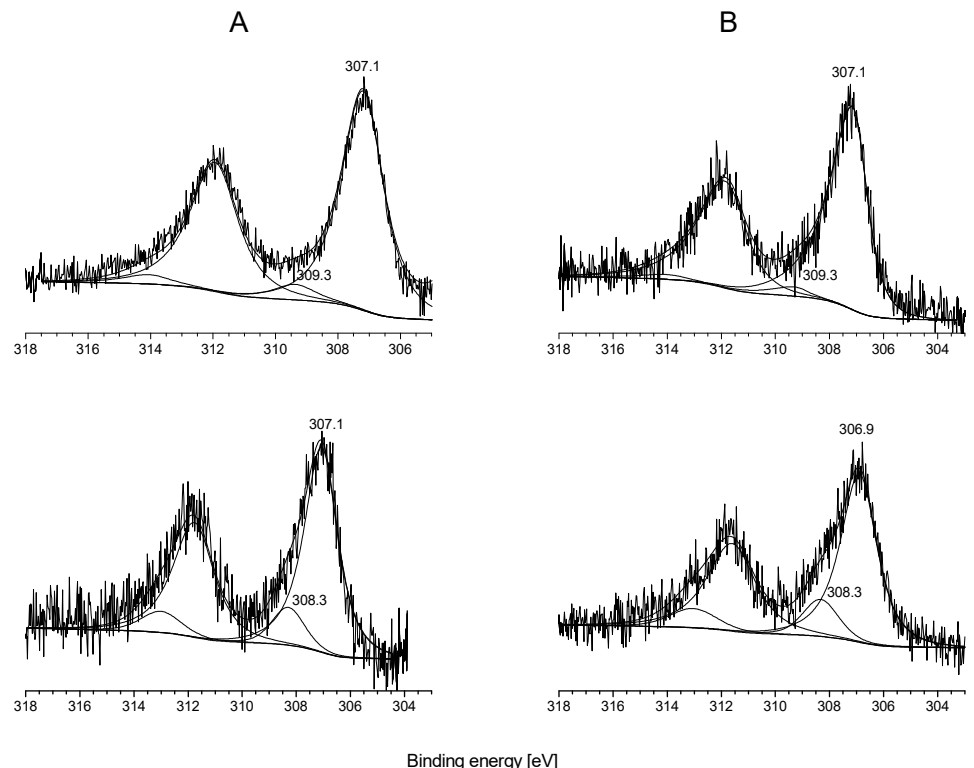

**Figure 5.** XP spectra of Rh 3d on titanate nanowire (**A**) and nanotube (**B**) with 1% Rh content (upper spectra) and 2% metal content (lower spectra). Reproduced from [70].

We demonstrate the effect of Rh-induced phase transformation in titanates and the influence of Rh nanoparticles and single $Rh^+$ ion stabilized in ion-exchange positions on the conversion and selectivity of the studied reactions. The ion-exchange possibility allows an atomic-scale distribution of metal cations in the titanate lattice. A suitable choice of the ionic form of the metal precursor can help significantly in increasing catalyst loading and maintaining a high catalyst dispersion. The ionic form of metals may increase the catalytic activity in cases where the redox mechanism is important. In some cases, we investigated the effect of co-adsorbed gold atom on the catalytic activity of Rh/titanates catalysts to study the special metal or ionic Rh and gold interactions.

## 2. Materials and Methods

Titanate nanowires (TiONW) and nanotubes (TiONT) were prepared by hydrothermal conversion of anatase $TiO_2$, as described previously [35,36,53]. After acid washing of titanates, most Na ion was replaced with hydrogen in this way and "H-form" titanate was obtained.

The prepared titanates can be characterized briefly; the outer diameter of the titanate nanotubes is 7–10 nm and their length is 50–170 nm, and they are composed of 4-6 wall layers. The diameter of their inner channel is typically 5 nm [35,43,53]. Titanate nanowires are the thermodynamically most stable form of sodium trititanate. Their diameter is 45–110 nm and their length is between 1.8 and 5 μm [36,43]. The specific surface area of titanate nanotubes is rather large (~185 $m^2g^{-1}$) due to their

readily accessible inner channel surface, whereas that of titanate nanowires is ~20 $m^2g^{-1}$. The BET surface area of Degussa $TiO_2$ P25 applied here was 50 $m^2g^{-1}$.

Rh/titanate nanocomposites were prepared by the impregnation method using $RhCl_3x3\ H_2O$ (Johnson Matthey) solutions to yield 1 and 2.5 wt% metal content. [33,43,87–90]. The samples were dried in air; finally, the catalysts were reduced in hydrogen atmosphere at 573 K for 1 hour. The characterization of the pristine and Rh-decorated titanates were made by XPS, HTEM, XRD and Raman spectrometry described in detail previously [43,70]. Bimetallic Au-Rh/titanates were prepared the same way [88–90]. Au, Rh and their coadsorbed layers with different composition were obtained by impregnation of the supports with the mixtures of calculated volumes of $HAuCl_4$ (Fluka) and $RhCl_3x3$ $H_2O$ (Johnson Matthey) solution to yield 1 wt % metal content.

For IR measurements, a Genesis (Mattson) spectrometer was applied. A BioRad FTS-135 FT-IR spectrometer supplied with a diffuse reflectance attachment was used for DRIFTS. The DRIFTS measurements were performed in an ultra-high vacuum system described previously [13,52,90]. The samples were pressed onto a Ta mesh. The mesh was placed at the bottom of a UHV sample manipulator. In total, 256 scans were registered at a spectral resolution of 2 $cm^{-1}$. A Whatman purge gas generator was used to purge the optical path.

The catalytic set up was described in more detail previously [20,52,89,90]. The reactions were carried out in a fixed bed continuous-flow reactor. The amount of catalyst used was usually about 0.1 g. The dead volume of the reactor was filled with quartz chips. The flow rate was usually 50 mL/min. Analysis of the product gases was performed with a Chrompack 9001 and Agilent 7890 gas chromatograph using Porapak QS columns. The products were detected simultaneously by TC and FI detectors with the help of a methanizer. The impregnated powders were dried in air at 383 K for 3 h. The final pre-treatment was at 573–600 K in hydrogen atmosphere. The $CO_2 + H_2$ reaction was studied at 493 K and 30,000 $h^{-1}$ mL $g^{-1}$ space velocity to achieve a relatively low conversion. The WGS reaction was carried out at 550 K while the ethanol decomposition was followed at 600 K. The amount and the activity of surface carbon formed in the catalytic reactions during 80 min were determined by temperature-programmed reduction (TPR). The catalyst was heated at a linear rate of 15 K/min hydrogen as carrier gas.

## 3. Effect of Titania Structure and Form of the Rh Metal on Heterogeneous Catalytic Reactions

### 3.1. CO_2 Hydrogenation on Titania and Titanate Supported Rh

The hydrogenation of $CO_2$ was studied extensively on titania ($TiO_2$) supported Rh [20–24,91–93] and the reaction was also investigated on titanate (TiONW and TiONT)-supported Rh recently [62,90,94]. In all cases, the supported Rh showed an excellent catalytic activity. The catalytic activities obtained on Rh/$TiO_2$, Rh/TiONW and Rh/TiONT are summarized in Table 2 and the effects of co-deposited Au are also displayed and compared with the results obtained on Au/$TiO_2$, Au/TiONW and Au/TiONT. The catalysts were pretreated by reduction with hydrogen at 573 K. At this temperature, the nanotube structure converted partially to anatase, while the Rh induced phase transformation from wire-like structure to $TiO_2$(B) phase also happened to an extent. In the case of Rh/TiONT, we mixed tube-like and nanoanatase composition, in the case Rh/TiONW, wire-like and $TiO_2$(B) structure co-exist. We note here that Degussa $TiO_2$ P25 has a mainly rutile structure. The main reaction product was $CH_4$ in all cases and minor CO formation was observed only on Rh/TiONT. Only traces of $C_2$ hydrocarbons were detected at 493 K. The methane conversions obtained at 493 K are displayed in Figure 6. H-form titanates were used always in the $CO_2$ hydrogenation experiments.

**Table 2.** Characteristic data for hydrogenation of carbon dioxide over Rh, Au, Au–Rh bimetallic clusters supported on titanate nanotubes, nanowires and $TiO_2$. The reaction temperature was 493 K.

| Catalyst | Amount of Adsorbed $H_2$ μmol/g | Conversion % | | $CH_4$ Formation Rate μmol/gs | | Turnover Number $s^{-1} \times 10^{-3}$ | $E_a$ kJ/mol | $\Sigma$ C μmol/g |
|---|---|---|---|---|---|---|---|---|
| | | in 5 min | in 80 min | in 5 min | in 80 min | in 80 min | in 80 min | |
| Rh/TiO$_2$ | 7.9 | 6.9 | 6.7 | 4.9 | 4.4 | 278 | 98.3 | 78.8 |
| Rh/TiONW | 7.5 | 8.9 | 4.5 | 6.6 | 3.2 | 213 | 96.5 | 121.5 |
| Rh/TiONT | 4.1 | 1.4 | 1 | 0.8 | 0.5 | 61 | 88.4 | 132.0 |
| Au-Rh/TiO$_2$ | 2.4 | 3.3 | 2.5 | 2.2 | 1.5 | 312 | 81.3 | 38.9 |
| Au-Rh/TiONW | 5.0 | 1.5 | 1.3 | 1.1 | 0.9 | 90 | 85.3 | 98.6 |
| Au-Rh/TiONT | 2.5 | 0.4 | 0.4 | 0.2 | 0.1 | 20 | 98.8 | 215.7 |
| Au/TiO$_2$ | 0 | 0.0006 | 0.0002 | $3.7 \times 10^{-4}$ | $1 \times 10^{-4}$ | - | - | 17.2 |
| Au/TiONW | 0 | 0.005 | 0.09 | $3.5 \times 10^{-3}$ | $6 \times 10^{-4}$ | - | - | - |
| Au/TiONT | 0 | 036 | 0.098 | $8.3 \times 10^{-4}$ | $2.1 \times 10^{-4}$ | - | - | 3.0 |

$E_a$ Activation energy for $CH_4$ formation. $\Sigma$ C Amount of surface carbon formed in the reaction at 493 K during 80 minutes.

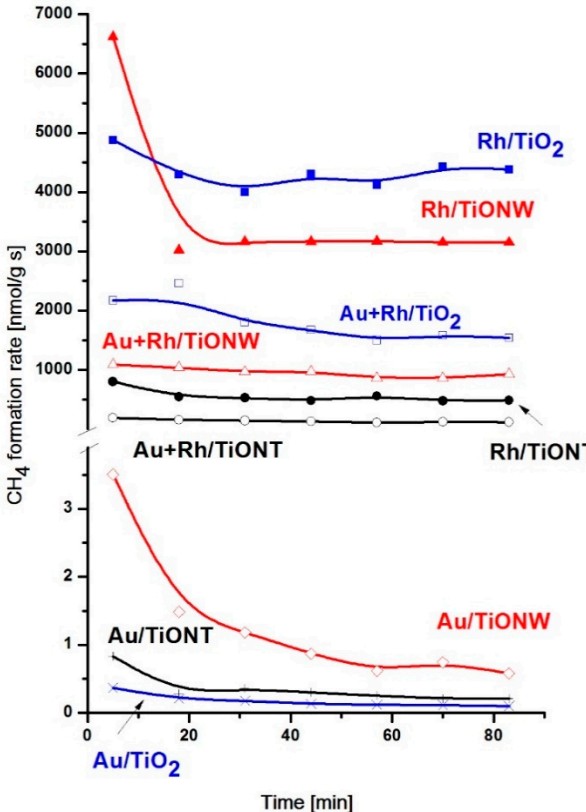

**Figure 6.** Rate of methane formation over Rh/TiO$_2$, Rh/TiONW, Rh/TiONT, Au–Rh/TiO$_2$, Au–Rh/TiONW, Au–Rh/TiONT, Au/TiO$_2$, Au/TiONW, Au/TiONT catalysts at 493 K.

The activity order of the supported Rh samples in the first minutes of the reaction decreased in the order Rh/TiONW > Rh/TiO$_2$ > Rh/TiONT. The conversion of CO$_2$ on Rh/TiONW decreased significantly in time, whereas in the other cases the CO$_2$ consumption was relatively steady. Rh/TiO$_2$ displayed a somewhat higher steady state activity. The differences in activity cannot be explained by different surface areas. We should consider several other factors. No doubt that the different titanate phases had a decisive role. It seems that the mixed tube-like nanostructured anatase composition of

nanotubes does not prefer the methanation of $CO_2$. There is a significant difference in the number of the available active sites evidenced by the $H_2$ dispersion measurements. This can be attributed to the deactivation of Rh-based active sites that resulted from the structural transformation of the nanotubes due to their thermal instability. On the other hand, the ratio of the number of active Rh nanocluster and $Rh^+$ could be hardly determined, but we suppose that the positively charged Rh successfully helped the activation of $CO_2$ and the further reaction of intermediates (see below). It is remarkable that there is higher tendency of carbon deposition on titanates comparing to TiO2 P25, which could better inhibit the reaction. The amount of deposited carbon decreased in the order of TiONT > TiONW > $TiO_2$, with the exception of supported Au samples.

A drastic decrease in conversion was experienced when the Au-Rh bimetallic samples were used as catalysts but the activity order of the samples remained the same. We should note that the supported Au samples has a very poor activity in $CO_2$ hydrogenation. When we discuss the catalytic behavior of bimetallic catalysts, we should consider that the Au-Rh interaction on titanates (Au-Rh/TiONW) produces a core-shell structure similar to the well defined $TiO_2$(110). In previous works, it was demonstrated with STM, XPS and LEIS measurements that Rh core–Au shell clusters can be formed on $TiO_2$(110) if Au is post deposited by physical vapor deposition (PVD) on the substrate containing Rh clusters [91–93,95–97]. The surface composition of Au–Rh clusters on titanate nanocomposite was also investigated by LEIS [89]. As Figure 7 demonstrates, the Rh LEIS intensity decreased dramatically with the increasing gold content. The most intriguing feature was observed in the 0.5% Au + 0.5% Rh case. In monometallic systems, gold and rhodium $He^+$ scattering signals appeared at 753 and 707 eV, respectively. On bimetallic nanocomposite, however, only the gold signal showed up (Figure 7).

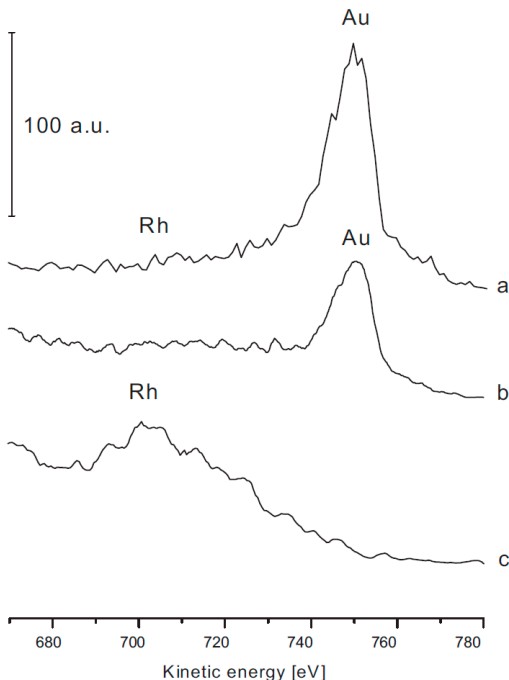

**Figure 7.** Low-energy ion scattering spectra (LEIS) of 1% Au/TiONW (b), 1% Rh/TiONW (c), 0.5% Au + 0.5% Rh/TiONW (a). Reproduced from [89].

If the Au completely and uniformly covers the Rh clusters (core-shell structure), it is plausible that adsorbed CO cannot occur (CO does not adsorb on Au surface at 300 K). Yet, strong CO bands appeared at 300 K at a pressure of 1.3 mbar. The peaks correspond to the linear form at 2070 cm$^{-1}$ and the twin CO mode at 2098 and 2033 cm$^{-1}$ (Figure 8B).

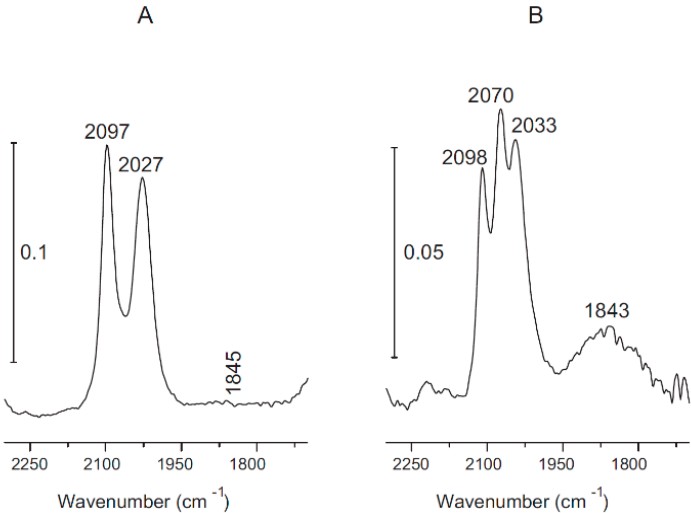

**Figure 8.** FTIR spectra of adsorbed CO at 300 K: (**A**) 1% Rh/TiONW and (**B**) 0.5% Au + 0.5% Rh/TiONW. Reproduced from [89].

Apparently, there is a contradiction between the results of LEIS and CO adsorption infrared experiments. There are no Rh atoms in the topmost layer (Figure 7), but adsorbed CO was detected by FTIR on this surface (Figure 8). This discrepancy can be explained by a CO-induced surface reconstruction. The adsorption of CO on Au–Rh clusters may promote the diffusion of Rh to the surface of the cluster. Similar phenomena were observed recently in other bimetallic systems on $TiO_2$(110) [98,99]. The adsorption of CO on bimetallic clusters can induce the diffusion of Rh to the surface from the core-shell structure. The CO may destroy the core-shell. Another possibility is that there is a continuous thermal fluctuation of Rh and Au atoms within the bimetallic clusters, and for short periods, Rh atoms can appear on the cluster surface on which the CO adsorption may occur [100,101]. In the case of $CO_2$ methanation on Au-Rh/TiONW, the activity loss can be explained by the distortion of the core-shell structure by reactants, as was discussed in the case of CO interaction. In any case, further studies are needed to understand the reactants-induced restructuring of Rh–Au clusters in detail.

The activation energy of $CO_2$ hydrogenation was calculated from the temperature dependence of the $CH_4$ formation rate at the steady state. The obtained 81–98 kJ/mol value is in a good agreement with previous results obtained for this Rh catalysized reaction [20]. It is important to note that the activation energies calculated on monometallic or bimetallic (Au-Rh) samples were almost identical (Table 2).

The infrared spectra registered in the DRIFT cell during $CO_2$ hydrogenation showed that on Rh/TiONW (Figure 9) and Rh/TiONT (Figure 10), an absorption band was present in the CO region from the beginning of the reaction at 2045 and 2049 $cm^{-1}$, respectively. The intensities and the positions of these bands did not change significantly during the catalytic reaction. On Rh/TiONW, absorptions at 1775–1765, 1628, 1557–1555, and 1379 $cm^{-1}$ were found (Figure 9). On Rh/TiONT, a shoulder was also observed at about 1960 $cm^{-1}$ and bands were detected at 1767, 1640 $cm^{-1}$ and 1568 $cm^{-1}$ (Figure 10). Similar spectral features were found when Au-Rh/TiONW and Au-Rh/TiONT were used as catalysts, but the intensities of the CO bands and the band at 1770 $cm^{-1}$ were weaker. On Rh/TiO$_2$, an intensive absorption was detected at 2049 $cm^{-1}$ and weak bands were observable at 1620 and 1570 $cm^{-1}$. There was no peak at ~1767 $cm^{-1}$.

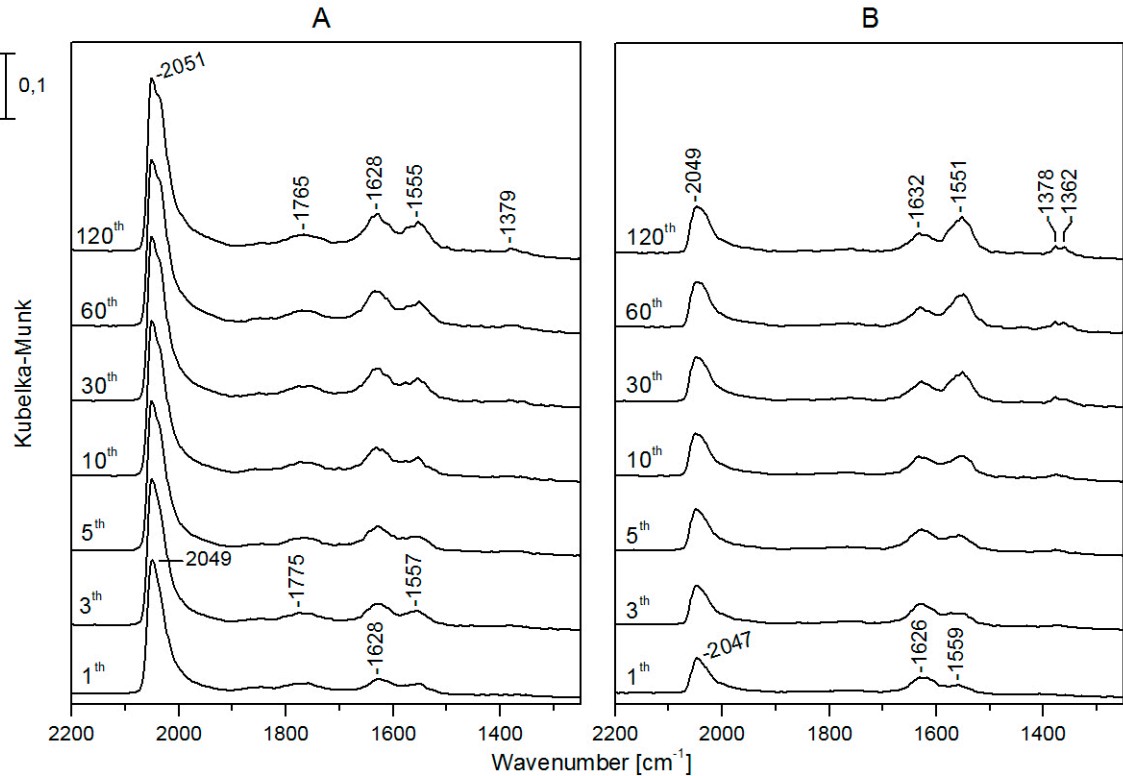

**Figure 9.** Infrared spectra registered during $CO_2$ + $H_2$ reaction at 493 K on Rh/TiONW (**A**) and Au-Rh/TiONW (**B**). The spectral labels indicate the time (in minutes) passed since the beginning of the reaction.

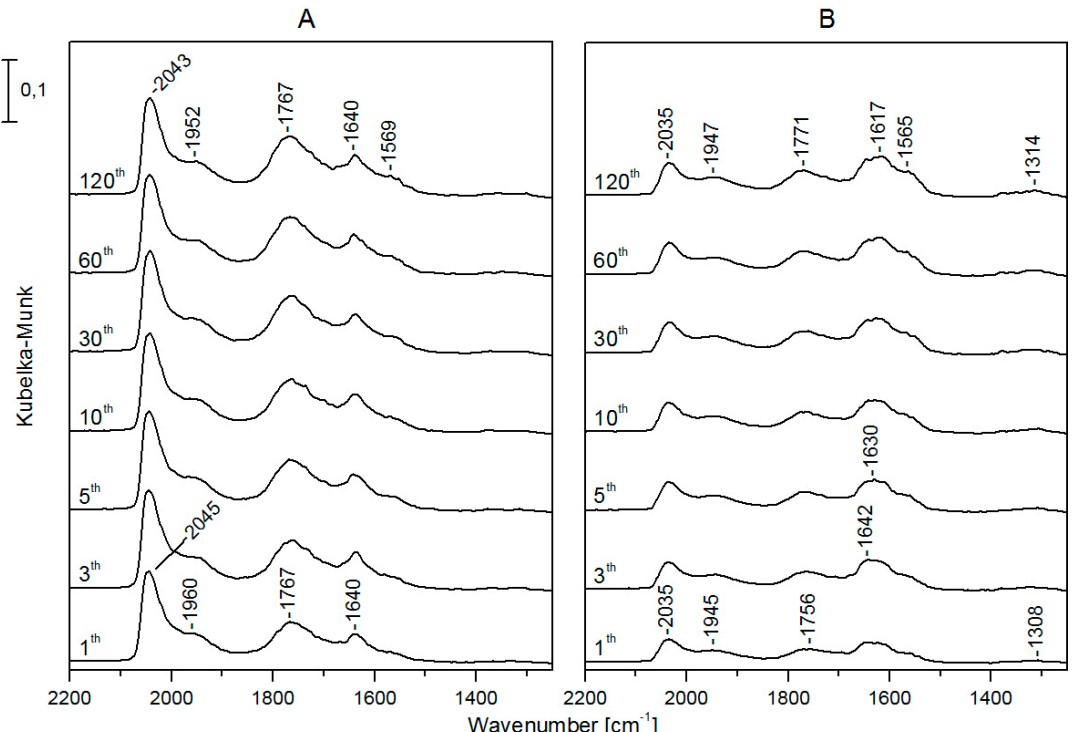

**Figure 10.** Infrared spectra registered during the $CO_2$ + $H_2$ reaction at 493 K on Rh/TiONT (**A**) and Au-Rh/TiONT (**B**). Spectral labels indicate the time (in minutes) passed since the beginning of the reaction.

The bands detected between 1550–1570 cm$^{-1}$ and 1379 cm$^{-1}$ could be assigned to the asymmetric and symmetric vibration of the OCO group of formate species, respectively [94,102–107]. The absorption found at about 1620 cm$^{-1}$ could be attributed to water formed in the reaction. The other bands below 1700 cm$^{-1}$ are due to different carbonates bonded to the supports [108].

The assignation of the band at ~1760 cm$^{-1}$, detected only on nanostructured titanate support is more complicated. This band was not observed on titania-supported Rh catalysts [20,91]. We could assign this band tentatively to formaldehyde of formic acid. However, the absorption band of the C=O group of formaldehyde adsorbed on Rh/TiO$_2$ appears at lower wavenumbers, at about 1727 cm$^{-1}$, and it forms at higher temperatures [109]. Although the vibration frequency of C=O groups in gaseous HCOOH is 1770 cm$^{-1}$ [110], our investigated feature cannot be assigned to this band because it remained observable even when the samples were flushed with He after the catalytic reaction. Low-frequency CO vibration (under 1790 cm$^{-1}$) has been observed in CO adsorption on Mn, La, Ce, Fe promoted Rh/SiO$_2$ catalysts [111–113]. The same feature appeared on Pt/zeolites during CO$_2$ hydrogenation [114]. It was suggested that Lewis acid sites caused the downward shift of the CO ligand vibration through the interaction between the Lewis acid and the oxygen atom of CO. The carbon atom of chemisorbed CO bonded to a Rh atom and its oxygen tilted to a metal ion. We are inclined to assign the investigated band at about 1770 cm$^{-1}$ to such tilted CO bonds to the Rh and interaction with an oxygen vacancy (Ti$^{3+}$) of the titanate support. When Rh was partially covered by gold, the intensity of this band decreased (Figures 9 and 10). The tilted CO configuration is illustrated in Figure 11.

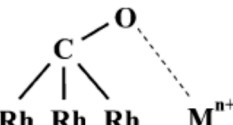

**Figure 11.** Schematic illustration of the tilted CO configuration on Rh/titanate catalysts. M$^{n+}$ represents a Ti$^{3+}$ site.

Taking into account the surface intermediates formed during the reaction (adsorbed CO and formate) and the reaction products (mainly methane and, to a lesser extent, CO), we propose that the hydrogenation of CO$_2$ proceeds via the reversed water gas shift reaction mechanism [115,116] and via Rh$^+$ and hydrogen assisted C-O cleavage in CO or H$_n$CO [20,93,107,113].

The CO$_2$ activation (formation of anionic radical) may proceed more easily on defect sites and on Rh$^+$ located in the ion-exchange position on titanates, and then it reacts with the adsorbed hydrogen atom:

$$CO_{2(a)}^{-*} + 2H_{(a)} \rightarrow H_nCO_{(a)} + OH_{(a)} \tag{1}$$

$$H_nCO_{(a)} + H_{(a)} \rightarrow CH_x + OH_{(a)} \tag{2}$$

$$CH_{3(a)} + H_{(a)} \rightarrow CH_{4(g)} \tag{3}$$

In parallel, a realistic CO formation route could be the decomposition of bidentate formate [101,103]:

$$HCOO_{(a)} \rightarrow CO_{(a)} + OH_{(a)} \rightarrow CO_{(g)} + H_2O_{(g)} \tag{4}$$

On the other hand, formate bonded close to the metal-oxide interface decomposes forming CH$_4$ [104,106]. The metallic Rh could deliver a sufficient amount of hot hydrogen atoms to rapture the C-O bond in formate species, the reaction of formate might be also catalyzed by positively charged Rh ion in titanate framework:

$$HCOO_{(a)} + 2H_{(a)} \rightarrow H_2COH_{(a)} \rightarrow H_2C_{(a)} + OH_{(a)} \tag{5}$$

$$H_2C_{(a)} + 2H_{(a)} \rightarrow CH_{4(g)} \tag{6}$$

Coke formation detected after reaction can be ascribed to the subsequent dehydrogenation of $CH_{2(a)}$.

In the Na-Rh/TiONT case, formic acid was also detected as a product [101]. It was suggested that it is formed as formate species react with rhodium hydride:

$$HCOO_{(a)} + H_{(a)} \rightarrow HCOOH_{(g)} \tag{7}$$

When Na-form titanates are used [94], the catalytic activity may be different. In the case of $CO_2$ hydrogenation, Yu et al. [117] compared the activity of Pt/TiONT and Pt/TiO$_2$ samples. In their test reaction, Pt/TiONT catalysts exhibited a higher activity than Pt/TiO$_2$. The authors explained the activity of Pt/TiONT with its higher $CO_2$ adsorption capacity originating from its higher surface area and nanotubular morphology. Moreover, some active superficial species were identified by in situ infrared studies during the reaction. It has been reported previously that the presence of alkali metals in solid catalysts can induce the dissociation of $CO_2$ [11,12,118]. In the case of TiONT synthesized by the hydrothermal method, the $Na^+$ contained in its structure could promote the effective dissociation of $CO_2$ on the catalyst surface; thus, the rate of $CO_2$ hydrogenation could be higher.

In summary, we may conclude that the structure of the support has a significant influence on the activity of titania and titanate-like catalysts in $CO_2$ hydrogenation. Rh/TiONW, which features a TiO$_2$(B) phase at the reaction temperature has a higher activity than Rh/TiONT, in which the anatase structure is dominant at the reaction temperature. The Degussa TiO$_2$, which has a mainly rutile structure, exhibited a somewhat higher steady state activity than Rh/TiONW, although the TOF values are almost the same. The ionic form of Rh located on titanate framework may contribute to the activation of $CO_2$ and further reaction formate intermediates formed during catalytic reaction.

### 3.2. CO + H$_2$O Reaction on Rh/TiONW, Rh/TiONT and Rh/TiO$_2$

Water gas, also known as synthesis gas, contains carbon monoxide (CO) and hydrogen ($H_2$). The water gas shift (WGS) reaction is the intermediate step used for CO reduction and hydrogen enrichment in synthesis gas [119,120]. Italian physicist Felice Fontana discovered the water gas shift reaction in 1780, but its actual importance was realized much later. The water gas synthesis reaction is an important process to produce CO-free hydrogen or to adjust the $H_2$/CO ratio [121]. Adjusting the $H_2$/CO ratio is especially required for downstream processes such as Fischer-Tropsch reactions and methanol synthesis [122].

Many catalysts were tested in the WGS reaction. According to the nature of the active component and the applied support, $CO_2$, $H_2$ and different hydrocarbons were formed [15,16]. Based on the kinetic results, two types of mechanisms were proposed [1]. In the oxidation-reduction, or regenerative mechanism of Rideal-Elay type, water oxidizes the surface and CO re-reduces the oxidized surface [123]. Others have suggested a bi-functional process where the CO adsorbed on the precious metal or the mixed metal oxide is oxidized by the support and then water fills the support oxygen vacancy [124]:

$$H_2O + * \leftrightarrow H_2 + O^* \tag{8}$$

$$CO + O^* \leftrightarrow CO_2 + * \tag{9}$$

where $*$ is an active site.

Another possible explanation is offered by the multi-step Langmuir–Hinshelwood type or "associative" mechanism where adsorbed or dissociated water forms reactive hydroxyl groups that combine with CO to produce a formate that decomposes to $CO_2$ and $H_2$. Others describe a bifunctional nature where CO adsorbed on the reduced metal migrates to react with hydroxyl groups in a

bond-making reaction to produce the formate intermediate [15,16,125,126]. The FTIR analysis has been commonly used to confirm the presence of the formate intermediate:

$$CO + * \leftrightarrow CO^* \tag{10}$$

$$H_2O + 2a \leftrightarrow H^* + OH^* \tag{11}$$

$$OH^* + CO^* \leftrightarrow HCOO^* + * \tag{12}$$

$$HCOO^* + * \leftrightarrow CO_2^* + H^* \tag{13}$$

$$CO_2^* \leftrightarrow CO_2 + * \tag{14}$$

$$2H^* \leftrightarrow H_2 + 2* \tag{15}$$

At higher pressures corresponding to industrial conditions, it was necessary to include steps covering the synthesis and hydrogenation of formate [127]. In this hydrogenation reaction, methanol was produced:

$$HCOO^* + H^* \leftrightarrow H_2COO^* \tag{16}$$

$$H_2COO^* + 4H^* \leftrightarrow CH_3OH_{(g)} + H_2O_{(g)} \tag{17}$$

The catalytic reaction of water with carbon monoxide to form hydrocarbons and carbon dioxide is known as the Kölbel–Engerhardt reaction [128]. Earlier, this reaction was investigated over supported rhodium [129,130]. The formation of methane appears to occur via the water gas shift reaction, followed by the hydrogenation of surface carbon:

$$CO + 2s \leftrightarrow C_s + O_{(a)} \tag{18}$$

$$H_2O + 2s \leftrightarrow OH_{(a)} + H_{(a)} \tag{19}$$

$$C_s + 4H_{(a)} \leftrightarrow CH_{4(g)} \tag{20}$$

The turnover frequency for this reaction was a function of the support type, resulting in the turnover frequency sequence at 600 K: $Rh/Al_2O_3 > Rh/Y$ zeolite $> Rh/SiO_2 > Rh/NaY$ zeolite [129].

The $CO + H_2O$ reaction was investigated on Rh catalysts using titanate and titania supports. The main products were hydrogen and carbon dioxide; only a very small amount of methane was formed on titania-like supports. The catalytic activity of $Rh/TiO_2$, Rh/TiONW and Rh/TiONT was tested and compared at 550 K. The conversion data are displayed in Figure 12. The $CO_2$ formation rates are shown in Figure 13. The $H_2$ formation rate exhibited a similar trend (not shown). The highest conversion was measured on 1% $Rh/TiO_2$ (Degussa P25). The steady-state activity of 1% Rh/TiONW was higher than on 1% Rh/TiONT. A similar trend was found when we compared the formation rates. Although the conversion and the formation rate were the highest on 1% $Rh/TiO_2$, the turnover frequencies showed no significant deviation (Table 3). The effect of Au additive was also investigated. The Au/titanates were not active in the reaction.

**Table 3.** Activity data obtained in the $CO + H_2O$ reaction over different catalysts in the quasi steady-state reaction.

| | D % | K % | WCO$_2$ nmol/gs | TOF CO$_2$ s$^{-1}$ |
|---|---|---|---|---|
| 1% Rh/TiO$_2$ | 36 | 11.4 | 1078.1 | $30.8*10^{-3}$ |
| 1% Rh/TiONT | 10 | 3.4 | 295.6 | $30.4*10^{-3}$ |
| 1% Rh/TiONW | 29 | 4.7 | 456.8 | $16.2*10^{-3}$ |
| 1% (Rh+Au)TiONW | 33 | 1.9 | 176.8 | $5.5*10^{-3}$ |

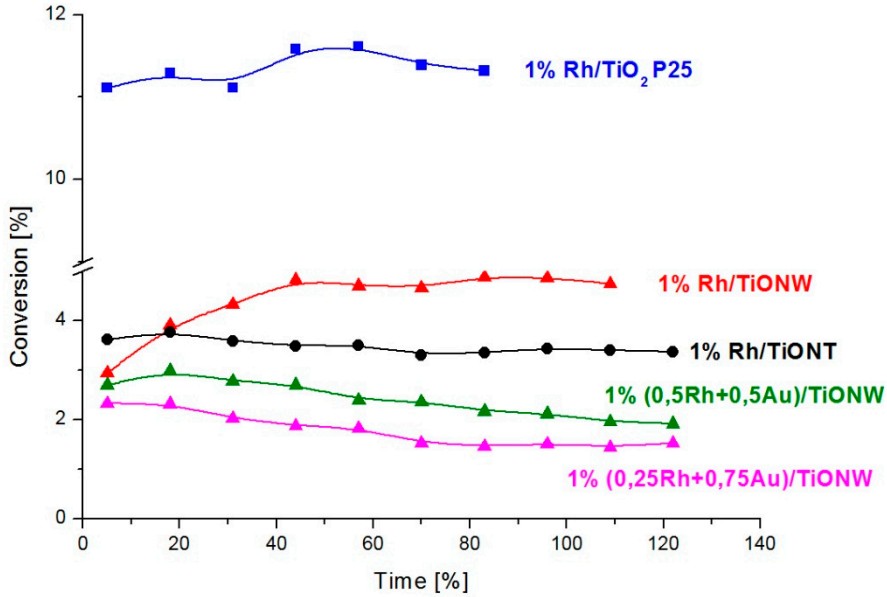

**Figure 12.** Conversion of the CO + $H_2O$ reaction over 1% $Rh/TiO_2$ (Degussa P25), 1% Rh/TiONW, 1% Rh/TiONT, 1% (0.25Rh + 0.75Au)/TiONW and over 1% (0.5Rh + 0.5Au)/TiONW. The reaction temperature was 550 K.

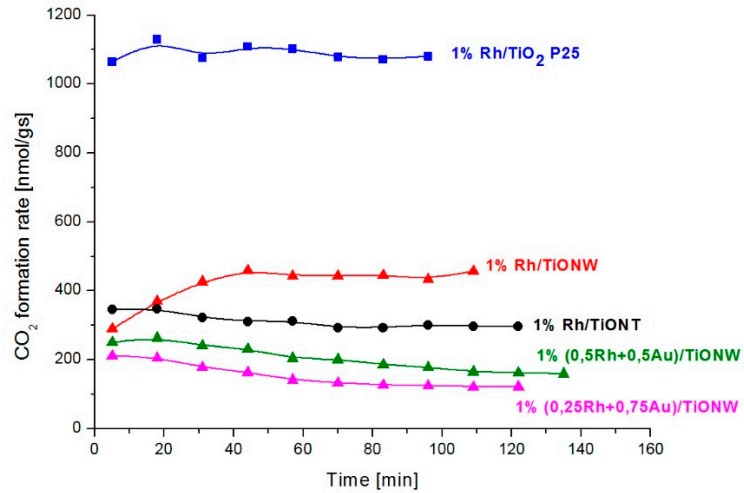

**Figure 13.** $CO_2$ formation rate in the CO + $H_2O$ reaction over different Rh catalysts at 550 K.

As the reaction products were almost exclusively $CO_2$ and hydrogen, we suggest that both the oxidation-reduction or regenerative mechanism of the Rideal-Elay type, and the multi-step Langmuir–Hinshelwood type or "associative" mechanism operate on titania- and titanate-supported Rh catalyst. The oxygen mobility, mainly in titanate supports, is high enough to subtract oxygen. Since formate intermediates are easily formed and stabilized on these supports or at the metal-support interface, the $CO_2$ formation can be explained by the decomposition of formate. As we discussed above, the Rh ion in the titanates may catalyze the reaction efficiently via the oxidation-reduction mechanism. Interestingly, the TOF values are the same for both $Rh/TiO_2$ and Rh/TiONT. This means that the number of Rh is higher on $Rh/TiO_2$ but the activities of Rh sites are the same.

The co-deposited gold suppressed the catalytic activity, as was detected in $CO_2$ hydrogenation, indicating that gold covers the active Rh sites. The observed smaller activity can be attributed to the Rh particles which were segregated from the Au-Rh core-shell cluster due to CO as a reaction partner of the WGS reaction. The reaction mixture destroys the core-shell structure, as discussed above.

*3.3. CH₅OH Decomposition*

The increasing demand for alternative energy sources has drawn great attention to $H_2$ production by ethanol transformation [17–19]. The use of ethanol is favored because it can be readily produced from renewable biomass. Supported noble metals are active catalysts in the transformation. Both the nature of the metal and the support determine the product distribution. Alumina-supported catalysts are very active at low temperatures in the dehydrogenation of ethanol to ethylene. At high temperatures, ethanol is converted into $H_2$, $CO$, $CO_2$ and $CH_4$. Rh was significantly more active and selective towards hydrogen formation than Ru, Pt and Pd [131–138]. Ethylene formation was not detected when a ceria-type support was used [132]. When titania was the support, aldehyde formation was also significant [134,135,137]. The surface chemistry of ethanol on Rh single crystal was also investigated in detail [139,140]. We now compare the performance of Rh supported on different nanostructured titanates (TiONW and TiONT) as the catalyst of the ethanol transformation reaction.

The 1% $Rh/TiO_2$ showed a high conversion at 603 K at the first stage of the reaction: at 5 min, the conversion reached 93%. With an increasing reaction time, the conversion dropped significantly (Figure 14). The steady state activity was obtained with ~ 40% conversion at around 120 min.

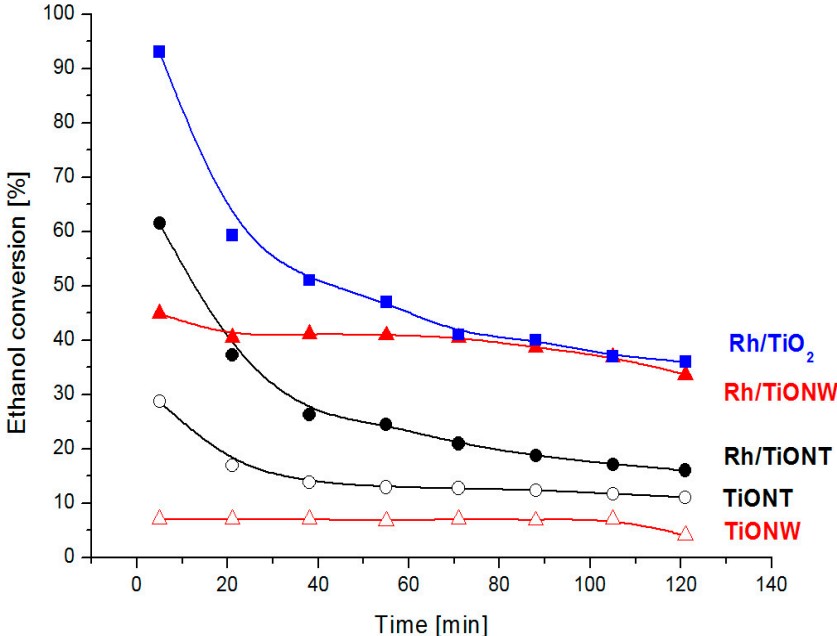

**Figure 14.** Ethanol conversion over titania and titanate supported Rh catalysts at 603 K.

When nanostructured titanates were applied, the activities were smaller. In the case of 1% Rh/TiONT, the initial conversion was around 60% and it decreased with time; the steady state activity was obtained at around 100 min with 20% conversion. Interestingly, the Rh/TiONW activity was higher than that of Rh/TiONT without any major activity decrease (Figure 14). The steady state activity was the same as that measured over $Rh/TiO_2$. Acetaldehyde, hydrogen, carbon monoxide, methane, ethylene and diethyl ether were the reaction products on all titania and titanate supported Rh catalysts. Very small amounts of aceton and acetic acid were also detectable occasionally. High acetaldehyde selectivity was detected in all cases, which indicates that the acetaldehyde forms via the oxidative dehydrogenation of ethanol as was assumed earlier on reducible oxide supports, including $TiO_2$ [133]. The relatively lower catalytic activity of Rh/TiONT can be explained by the tube structure; the diameter of the tube prevents the ethanol from reaching the active sites inside the tube.

The selectivity data at 5 and 105 min of the reaction are presented in Table 4. On $Rh/TiO_2$, acetaldehyde selectivity increased with time and hydrogen selectivity decreased. This decrease was less pronounced on both Rh/TiONW and Rh/TiONT where the hydrogen selectivity at steady state

activity was ~10%. This hydrogen selectivity value is significantly higher than that on Rh/TiO$_2$. The ethylene and diethyl ether selectivity was low in all cases.

**Table 4.** Conversion and selectivity data measured in the ethanol transformation reaction on titania- and titanate-supported Rh catalysts.

| | Conversion % | | Selectivity % | | | | | | | |
|---|---|---|---|---|---|---|---|---|---|---|
| | | | C$_2$H$_4$ | | C$_2$H$_4$O | | H$_2$ | | (C$_2$H$_5$)O | |
| min | 5. | 105. | 5. | 105. | 5. | 105. | 5. | 105. | 5. | 105. |
| Rh/TiO$_2$ | 93 | 37 | 6.9 | 1.8 | 77.8 | 95.7 | 13.6 | 5.8 | 7 | 0.1 |
| Rh/TiONW | 44 | 36 | 0.4 | 0.2 | 86.6 | 94.7 | 12.5 | 10.8 | 3.7 | 0.6 |
| Rh/TiONT | 61 | 17 | 0.4 | 0.3 | 78.0 | 91.3 | 25.0 | 10.6 | 1.0 | 3.6 |
| TiONW | 1 | 7.1 | 3.2 | 3.4 | 66.7 | 66.1 | 5.0 | 0 | 26.0 | 28.0 |
| TiONT | 28.7 | 11.7 | 23 | 0.8 | 93 | 96.0 | 6.0 | 0 | 1.0 | 2.4 |

Infrared spectroscopy is a useful tool to identify surface intermediates formed during a catalytic reaction. An in situ DRIFTS study was performed in the presence of a reaction mixture at different temperatures. The spectra were qualitatively the same on all supported catalysts. Figure 15 shows the typical spectra obtained on Rh/TiOW. At 300 K absorption bands were observed at 2974, 2929 and 2875 cm$^{-1}$ in the C–H stretching region. In the low-frequency range, absorption bands were detected at 1448–1450 and 1383–1384 cm$^{-1}$ and both could be assigned as the symmetric and asymmetric CH$_3$ vibrations of ethanol/ethoxide. The bands observed at 1140–1144, 1124–1121, 1069–1074, 1045–1047 cm$^{-1}$ could be attributed to νC–O and νC–C vibrations of monodentate and bidentate ethoxide species [132–134]. The intensities of these bands decreased as the temperature increased but dramatic changes were not detected. At ~1560 cm$^{-1}$, a small intensity peak appeared at 423 K, probably due to acetate species ($\nu_{as}$O-C-O). This band appears with a higher intensity on Al$_2$O$_3$ and CeO$_2$ supports [132–134].

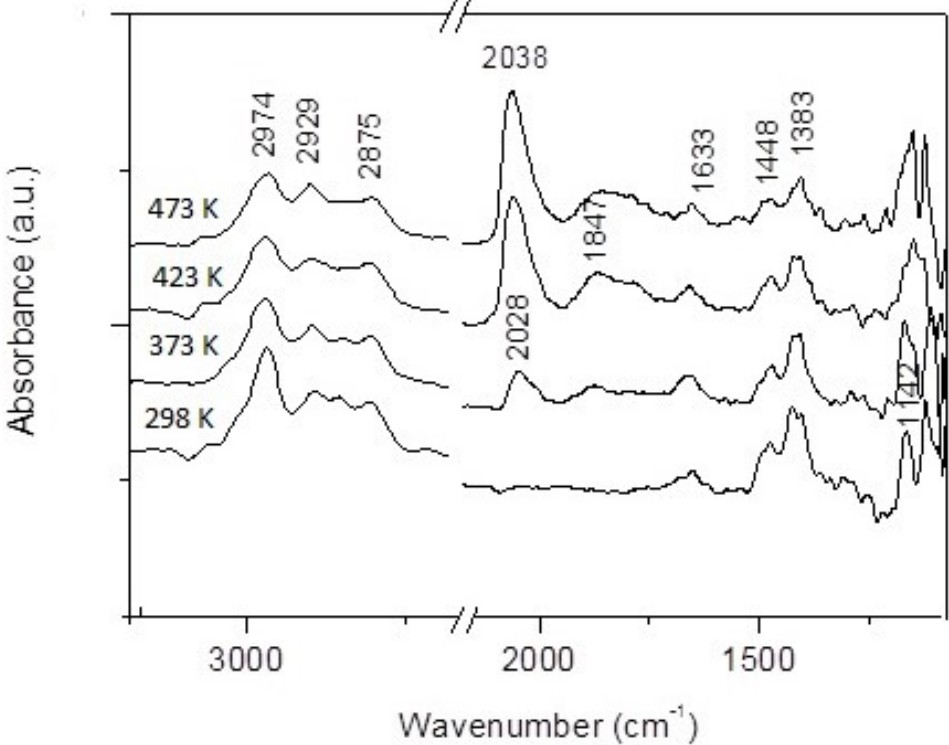

**Figure 15.** DRIFT spectra obtained on Rh/TiONW during ethanol decomposition at different temperatures.

Apart from the ethanol/ethoxide bands, a peak at 1633 cm$^{-1}$ started developing at 297 K. Its intensity increased slightly with temperature. This band is present in many cases in ethanol decomposition, oxidation and ethanol steam reformation [141,142]. It was attributed to the $\nu$C–O vibration in acetyl (CH$_3$CO) species. When the Rh/TiO$_2$ was treated with ethanol above 373 K, new peaks were detected at 2028 and 1847 cm$^{-1}$. The intensities of these bands increased by increasing the temperature and the peak observed at 2028 cm$^{-1}$ at room temperature shifted to higher wave numbers. The absorbance at 1847 cm$^{-1}$ could be attributed to bridge bonded CO on Rh sites [77,91]. The feature of the adsorbed CO detected at 2028–2038 cm$^{-1}$ is significantly differing from that depicted in Figure 4. This discrepancy could be explained by the Rh carbonyl hydride formation (H–Rh–CO) [143]. The adsorbed CO desorbs in a desorption rate-limiting step.

Taking into account the reaction products and the surface intermediates formed during the reaction, we offer the following reaction steps which were proved in many cases. After the initial step, the formation of ethoxide, further dehydrogenation occurs:

$$C_2H_5OH_{(a)} \rightarrow C_2H_5O_{(a)} + H_{(a)} \tag{21}$$

$$C_2H_5O_{(a)} \rightarrow CH_3CHO_{(a)} + H_{(a)} \tag{22}$$

The acetaldehyde mostly desorbs as reaction product and a smaller fraction decomposes forming acetyl:

$$CH_3CHO_{(a)} \rightarrow CH_3CO_{(a)} + H_{a)} \tag{23}$$

In the acetyl species, the C–C rupture easily happens, forming adsorbed CO and methyl radicals. Rh is considered to be among the best metals for hydrogen production due to its high activity for C–C cleavage in ethanol. The methyl radical either decomposes further or reacts with hydrogen and finally methane is produced. The adsorbed hydrogen recombines into H$_2$. As the hydrogen selectivity was two times higher on both Rh/TiONW and Rh/TiONT than on TiO$_2$ P25, the decomposition of acetyl intermediate could be catalyzed by Rh$^+$, which is absent on commercial TiO$_2$:

$$CH_3CO_{(a)} \rightarrow CH_{3(a)} + CO_{(a)} \tag{24}$$

$$CH_{3(a)} + H_{(a)} \rightarrow CH_4 \tag{25}$$

$$CH_{3(a)} \rightarrow CH_{(a)} + H_2 \tag{26}$$

During ethanol decomposition, very small amounts of ethylene and diethyl ether were also detected. In this process, the first step involves the formation of diethyl ether because of intermolecular dehydrogenation between two ethanol molecules [144]:

$$2C_2H_5OH_{(a)} \rightarrow C_2H_5\text{-}O\text{-}C_2H_{5(a)} + 2OH_{(a)} \tag{27}$$

This is followed by a second dehydrogenation of diethyl ether to ethylene:

$$C_2H_5\text{-}O\text{-}C_2H_{5(a)} \rightarrow 2C2H4 + 2OH_{(a)} \tag{28}$$

Summing up, we may conclude that the structure of nanotitanates influences the steady state conversion of ethanol to some extent. The conversion was equal on Rh/TiO$_2$ and Rh/TiONW, but lower on Rh/TiONT where the decomposition may sterically hindered by the tubular structure of TiONT. In all cases, acetaldehyde was the main product. It is remarkable that on Rh supported on nanostructured titanates, the hydrogen selectivity was double the steady state conditions. IR studies verified that the ethoxide dehydrogenates and most of the acetaldehyde desorbs as product but some of it dehydrogenates further and C–C bond rupture occurs via the acetyl surface intermediate.

## 4. Summary

Rh accelerated the transformation of titanate nanotubes to anatase. The strong interaction between Rh (in ionic form, discussed above) and titanate structure plays an important role in catalytic effects. In addition to this interaction, the structure of titanates moderates the catalytic activity. Rh-decorated nanowires transform into the $TiO_2(B)$ phase and the nanotubes transform to anatase. The Rh-assisted processes started from 550 K where the titanate-supported Rh catalysts exhibited a remarkable catalytic effect in $CO_2$ hydrogenation, $CO + H_2O$ reaction and in ethanol decomposition. In all cases, the use of $TiO_2$ P25 (which contains mostly rutile phase) performed the highest activity at steady-state conditions.

The activity order of the supported Rh samples in the first minutes of the reaction decreased in the order $Rh/TiONW > Rh/TiO_2 > Rh/TiONT$ in $CO_2$ hydrogenation. $Rh/TiO_2$ displayed the highest steady-state activity. The conversion of $CO_2$ on Rh/TiONW decreased significantly in time but in the other cases, the $CO_2$ consumption was relatively constant. The reaction ran in all cases via the formation of a formate intermediate in $CO_2$ methanation. On both nanostructured titanates, "tilted CO" species were also formed, which facilitates H-assisted C-O bond rupture. Positively charged Rh ion may help the activation of $CO_2$ and accelerate the further decomposition of formate intermediates yielding $CH_4$.

In the $CO + H_2O$ reaction, the highest conversion was obtained on $Rh/TiO_2$ Degussa P25 catalyst. It is very interesting that the turnover frequencies obtained in the case of Rh/TiONT, Rh/TiONW and $Rh/TiO_2$ were almost the same. From this comparison, we may suggest that the number of Rh is higher on $Rh/TiO_2$ but the activities of Rh sites are the same. As the reaction products were almost exclusively $CO_2$ and hydrogen, we suggest that both oxidation-reduction or regenerative mechanism of the Rideal-Elay type, and a multi-step Langmuir–Hinshelwood type or "associative" mechanism operate. The oxygen mobility, mainly in titanate supports, is high enough to subtract oxygen for the reaction.

In ethanol decomposition, the $Rh/TiO_2$ and Rh/TiONW exhibited an equal steady-state activity. The relatively lower catalytic activity in the case of Rh/TiONT can be explained by the tube structure: the diameter of the tube inhibits the diffusion the ethanol reaching the active sites inside the tube. In all cases, acetaldehyde was the main product. A small fraction of acetaldehyde transforms to acetyl intermediates with help of $Rh^+$. The C–C breaking in acetyl species leads to the formation of hydrogen. It is remarkable that on nanostructured titanates, the hydrogen selectivity was two times higher than on $Rh/TiO_2$ at steady-state conditions.

The $TiO_2$ (Degussa P25) contains a dominantly rutile structure (rutile/anatase ratio is roughly 75/25). The Rh/TiONW performs a significant $\beta$-$TiO_2$ phase at reaction conditions. The Rh/TiONT transforms mainly to anatase structure. From the comparison of conversion data of the studied reactions, we conclude that the rutile structure shows the highest activity. In steady-state conditions, the rutile structure exhibits almost the same activity than the $TiO_2(B)$ phase in $CO_2$ methanation and ethanol decomposition. In the case of Rh/TiONT where the anatase phase is dominant, less activity was measured in all three reactions.

The tube morphology in titanate nanotubes case may also influence the catalytic activity; the diffusion of reactants and the products could be different on the outside and the inside of the tubes.

The gold adatoms decrease the activity of Rh due to the formation of core-shell structure where the gold is on top. The observed activity can be attributed to the partial disruption of the structure by the reactants and the lower activity of the remaining Au nanoparticles in the topmost layer.

**Author Contributions:** All authors contributed to write the review. All authors have read and agreed to the published version of the manuscript.

**Funding:** This research received no external funding.

**Acknowledgments:** The authors wish to thank Péter Pusztai and Kornelia Baán for executing the electron microscopic and infrared measurements and András Erdőhelyi for fruitful discussions. Financial support of this work by the National Research Development and Innovation Office through grants GINOP-2.3.2-15-2016-00013

and NKFIH OTKA K120115 (Zoltán Kónya), K126065 (Ákos Kukovecz) and PD 120877 (András Sápi) are gratefully acknowledged.

**Conflicts of Interest:** The authors declare no conflict of interest.

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
