# Peer review of "Rh-induced Support Transformation and Rh Incorporation in Titanate Structures and Their Influence on Catalytic Activity"

_catalysts, doi:10.3390/catal10020212_

Round 1

Reviewer 1 Report

Review of “Rh-induced Support Transformation Phenomena in Titania and Titanate Catalysts and Their Influence on the Catalytic Activity [...]”

The paper by Konya and coworkers aims to describe the correlation between support transformation phenomena in titania and titanate catalysts and catalytic activity. I have found this paper to be well written and well referenced all throughout. However, prior to publication, I will recommend a major revision, as the current version has some global issues:

I would recommend a restructuring of the review text. Up until section 4 the text reads as an enumeration of surface-characterizations of various materials and just phenomenological observations. Perhaps moving section 4 as section 1, at the start of the text and then tying in the catalytic behavior to each of the mechanisms and processes. Section 4, where the catalytic activities are described is disjointed from the rest of the text. The reviewer was looking for causality relationships between the observed phenomena and catalytic activity from a mechanistic context. Unfortunately, such causality is loosely identified and needs to be strengthened. It seems that in some case these transformation phenomena don’t actually matter, but the why is not identified. The text of the proof I am reading has inconsistent capitalization of the title. In addition, the “v”s and the “y”s look exactly the same. It looks like the template truncated the bottom of the text. This needs to be addressed.

Overall, the review has the potential to be a meaningful contribution to the literature and we look forward to seeing it in print in the near future.

Author Response

We thank the Reviewer for the positive evaluation and valuable suggestions.

As the Reviewer requested we reconstructed the review text. The catalytic results section moved just after the “Materials and Methods”. So the catalytic behavior is not disjoined from the characterization part. In the revised version we tried to strengthen the causality between structure, surface phenomena, character of Rh and catalytic activity. All changes in the Introduction and catalytic parts are indicated by red. Yes, the Reviewer is right, the transformation phenomena is not enough to explain the all affects. It is one of the important factor. Rh+ formation and stabilization in the ion-exchange position and its effect on the reaction of intermediates are also significant. The number of active Rh nanoparticles and the Rh+ could be different on each titanates. It is important that Rh ion formation can be excluded on TiO2 P25 (there is not ion-exhange positions). The contributions of each important parameter cannot be estimated exactly at this moment. Interestingly, the differences in BET surface does not play significant role. The changes and addenda are made in Abstract, Introduction and in the presentation of the catalytic results. They are labeled by red. The title is changed. I hope it is much better in present form.

Reviewer 2 Report

The paper presents extensive, well thought-out characterization data of Rh-TiO2 catalyst for reforming related chemistry. While activities reported are poor, relative to a standard P25 catalyst, the structural information from use of nanotubes is of high value.  This paper may be suitable for publication if revised for better clarity and better labeling/explanations of titanium oxide phases.

Major

The abstract, introduction, and concussion do not do enough to establish how the various reaction chemistries tested are The paper fails to properly identify the characteristic diffraction patterns and spectra peaks of the phases that the authors discus. This should be done clearly on all such figures (and preferable in figure captions as well.  Without this it s very difficult to follow the authors arguments about the structure effect. There authors would also do well to better describe the less common phases described such as “titanates”. Also of note “Rh decorated nanowires  transform into the β-TiO2 structure, whereas their pristine counterparts recrystallize into anatase.”  β-TiO2 is anatase, making this comment very odd. Perhaps the authors mean TiO2(B) or some other phase?

Minor

The English is generally excellent.  Some minor errors and awkward word choices are exhibited throughout (Particularly the abstract)

The title is overly long.  The keywords/abstract may also benefit from use of more industrial type terminology like water-gas-shift and methanation (CO2  hydrogenation) to improve the searchability of the paper.

Author Response

We thank the Reviewer for the positive evaluation and valuable suggestions.

We have made changes in Abstract, Introduction and in the presentation of the catalytic results. We strengthened the causality between the characterization of catalysts and reaction chemistries in the revised version. Besides the phase transformation the chemical nature of Rh nanoparticles and ions are also important for catalytic point of view. The changes are signed by red. In the revised version, the reflections on the Figures are marked with black point and it is indicated on the Figures Captions in the revised version. We used TiO2(B) instead of β-TiO2. Thank you this remark. We modified the title and the keywords. These and the modified Abstract are more benefit for industrial type terminology.

Round 2

Reviewer 1 Report

The authors have satisfactorily addressed the prior reviewer comments. We can therefore recommend for publication after some minor spell checking corrections.

Author Response

Answer to reviewer:

Thank you for your suggestion. We have made English spelling.

Corrections are marked by red.

This manuscript is a resubmission of an earlier submission. The following is a list of the peer review reports and author responses from that submission.

Round 1

Reviewer 1 Report

Figure 1, Figure 2, Figure 3, Figure 4, Figure 5 and Figure 7 are a copy past of the article "ref 58"; all the discussion between line 168 to line 363 is a copy past of a the article "ref 58". It is not possible to just copy and use what was already published in other journal, even if it is from the same authors.

So this is not an original work. 

the authors have to re-write the article. they can refer to their other article but they have to cut down the discussion dealing with the characterization to a couple of lines.

Reviewer 2 Report

This manuscript summarizes the Rh-induced transformation of titanate nanowires and nanotubes and the surface characterization of the titanate supported Rh-based catalysts. Overall, this is a comprehensive work, however, I found some improper arrangement of the context in the manuscript. Therefore, I would suggest a minor revision and my comments are as follows:

The subtitle of Section 3 is Results and Discussion, which I think is not reasonable. The whole section (Page 4~16) focused on the review of literature information on the structure and properties of the Rh-decorated titanates. All the data, figure, and tables were taken from the previous publications and there are no new data from the present work. Therefore, this section should be named as a literature review instead. Owing to the same reasons listed above, the experimental section should be modified as well. The authors presented a very detailed description of the synthesis procedure and characterization techniques, such as XRD, XPS, IR, Raman, TPD, GTA, TEM, etc. However, most of these characterization techniques were from the literature review part, and only IR was employed in the present work. Therefore, the authors should remove those irrelevant experimental descriptions and provide describe the procedures according to the present work. A more detailed description of the catalytic performance tests should be provided in the experimental section, such as pretreatment, space velocity, temperature ramping rate, etc. I found the authors only reviewed the publications by their own group, which have already been well documented in their previous review work. However, titanate as supports materials for metal catalysts has been widely studied in the past few years. Those work by other groups should also be included at least in the introduction section.

Reviewer 3 Report

Review of “Rh-induced Support Transformation Phenomena in Titania and Titanate Catalysts and Their Influence on the Catalytic Activity” by Kónya and coworkers

Journal: Catalysts

Manuscript ID: catalysts-619434

This is a review of a paper by Kónya and coworkers that describes the influence of support transformation phenomena in titania and titanate catalysts in the presence of Rhodium. the paper was difficult to read and the hypothesis of the work cannot clearly be identified. It was also not clear whether the paper is in fact a review as listed in the portal and in the manuscript or whether it is a report of data collected by the authors. If the latter is the case, then we would encourage the authors to include a supporting information describing their experimental methods in detail.

Unfortunately as it stands the work presents phenomenological series of observations in no clear conclusions can be drawn. The reasoning presented is not clear and again the hypothesis that is being tested needs to be clarified. A significant rewrite of this manuscript would be beneficial for its successful publication.

In addition, we would encourage the authors to amend the title of their paper to reflect the type of catalysis that is currently being tested. We would also encourage the authors to Additional details as to why they think the changes in these catalytic supports are improving or hindering catalytic processes.

Overall the paper has the potential to be an important report in the field of heterogeneous catalysis and we would be happy to review a revised version at a later time.

There are also some formatting issues in the acknowledgements section, where names various acknowledged contributors have been compromised by the conversion to PDF.